

# Prediction of stock price direction using the LASSO-LSTM model combines technical indicators and financial sentiment analysis

Junwen Yang[1], Yunmin Wang[2] and Xiang Li[3]

[1] Chongqing Technology and Business University, School of Mathematics and Statistics, Chongqing, China
[2] College of Big Data Statistics, Guizhou University of Finance and Economic, Guiyang, Guizhou, China
[3] Chongqing Technology and Business University, Economic Research Center of the Upper Yangtze River, Chongqing, China

## ABSTRACT

Correctly predicting the stock price movement direction is of immense importance in the financial market. In recent years, with the expansion of dimension and volume in data, the nonstationary and nonlinear characters in finance data make it difficult to predict stock movement accurately. In this article, we propose a methodology that combines technical analysis and sentiment analysis to construct predictor variables and then apply the improved LASSO-LASSO to forecast stock direction. First, the financial textual content and stock historical transaction data are crawled from websites. Then transfer learning Finbert is used to recognize the emotion of textual data and the TTR package is taken to calculate the technical indicators based on historical price data. To eliminate the multi-collinearity of predictor variables after combination, we improve the long short-term memory neural network (LSTM) model with the Absolute Shrinkage and Selection Operator (LASSO). In predict phase, we apply the variables screened as the input vector to train the LASSO-LSTM model. To evaluate the model performance, we compare the LASSO-LSTM and baseline models on accuracy and robustness metrics. In addition, we introduce the Wilcoxon signed rank test to evaluate the difference in results. The experiment result proves that the LASSO-LSTM with technical and sentiment indicators has an average 8.53% accuracy improvement than standard LSTM. Consequently, this study proves that utilizing historical transactions and financial sentiment data can capture critical information affecting stock movement. Also, effective variable selection can retain the key variables and improve the model prediction performance.

# INTRODUCTION

Recently, the stock market prediction methods have attracted wide attention in academia and business. Some researchers suggest that stock price movement direction can not be predicted and propose the theories, such as the Efficient Market Hypothesis and the

Corresponding author
Yunmin Wang, dumbleduo@163.com

Random Walk Hypothesis (*Fama, 1970*; *Fama, 1995*). Proponents of the Efficient Market Hypothesis believe that all information in the stock market is reflected in the stock prices and availability for all traders. Therefore, investors cannot obtain excess profits by analyzing past stock prices (*Borges, 2010*).

However, some scholars applied empirical analysis to prove that the stock market is not effective completely and can be predicted (*Eom et al., 2008*; *Raza, 2017*). In this context, researchers propose many methods for predicting the stock market, such as (1) Technical Analysis; (2) Fundamental Analysis and (3) Machine Learning. Fundamental analysis assumes that the country's economic situation can reflect the development situation of a company and affect its stock price. Predictor indicators in the fundamental analysis are usually macroeconomic time series indicators, such as GDP, interest rate, currency exchange rate and customer price index (*Boyacioglu & Avci, 2010*). Technical analysis is the most commonly used forecasting method (*Cavalcante et al., 2016*). Technical analysts utilize stock price, trading volume, breadth and historical behaviour of trading activities to evaluate the stock status and predict price trends or trading signals. The assumption of technical analysis is that all market information, such as investor sentiment and macroeconomic conditions, is reflected in stock prices. Therefore, analysis based on information such as stock price and trading volume can predict the movement direction of the stock market (*Neely et al., 2014*; *Edwards, Magee & Bassetti, 2018*; *Agrawal et al., 2022*).

Each stock price forecasting method has different advantages and defects. Generally, machine learning models can learn the relationship between predictor variables and stock movement direction in historical data. Due to its effectiveness and extensibility, machine learning approaches could be applied to most problems in economic domains. Compared to traditional statistics methodologies and econometric models, machine learning models have better prediction performance and robustness (*Patel et al., 2015*; *Bustos & Pomares-Quimbaya, 2020*; *Shobana & Umamaheswari, 2021*). Commonly used machine learning methodology includes artificial neural network (ANN), linear discriminant analysis (LDA), random forest (RF) and support vector machine (SVM). SVM uses the Hinge loss function to calculate empirical risk and adds regularization terms to the solving system to optimize structural risk, which is a classifier with good robustness (*Chen & Hao, 2017*). RF is an ensemble model composed of multiple decision tree models, which can effectively avoid the over-fitting situation (*Ibrahim, Kashef & Corrigan, 2021*). However, these methods cannot capture the time series characters in stock data and have poor performance in dealing with high-dimensions data. Moreover, the training speed of the SVM model is slow and it is difficult to process in large-scale data samples, RF is built based on the decision tree model. When the number of decision trees is large, the training of the random forest model will consume a lot of computing power and time, which will lead to the low operating efficiency of the model. Moreover, these methods cannot capture the time series characters in stock data and have poor performance in dealing with high-dimensions data (*Nikou, Mansourfar & Bagherzadeh, 2019*).

Deep learning models can solve problems that exist in massive data sets, such as high dimensionality, high complexity and data noise, which are difficult for traditional machine learning algorithms to handle (*Thakkar & Chaudhari, 2021*). Recurrent Neural Network

(RNN) adopts the self-feedback neurons and constructs a directed loop network. Compared with the traditional DNN structure, the core idea of RNN is to decompose the information into a series, all recurrent units are connected and could pass the results of the previous moment to the next moment. This structure enables RNN to have a short memory function and could deal with time-series data (*Sherstinsky, 2020*). However, the RNN model has a long-term dependency problem, the more distant nodes from the current node have less and less influence on the processing of the current node. The sigmoid and the Tanh activation function in the RNN model can not process the important information of data when dealing with the large gap input vectors. Long Short-Term Memory (LSTM) is a variant of RNN, which can solve the problem of gradient explosion and gradient disappearance caused by time length (*Yu et al., 2019*). Compared to traditional machine learning methods, the LSTM model can effectively process the information in time series data and implement efficient prediction. The deep learning model is applied widely in numerous fields and yields good results. *Yu & Yan (2020)* use the time series phase-space reconstruction to process stock data and predict the stock price based on the deep neural networks (DNNs) model; *Su, Xie & Han (2021)* propose the RFG-LSTM model to predict stock up and down, which utilizes the rectified forgetting gate to improve the traditional LSTM model and reduces the computational complexity.

A wide range of indicators have been applied to predict the movement of stock, and the most commonly used are time series stock prices, technical indicators and finance text data. *Dai, Zhu & Kang (2021)* apply the wavelet technology to stock data de-noising and obtain the technical indicators, which can reflect the market behavior and stock trend dynamically. With the popularization of the natural language process (NLP) technique, massive financial textual data have been analyzed to predict the movement of stock, such as media articles (*Garcia, 2013*), finance news, and social media comments (*Renault, 2020*). Sentiment analysis usually extracts the characters from textual content and utilizes sentiment indicators to predict the trend of stocks (*Nam & Seong, 2019*; *Zolfaghari & Gholami, 2021*). In this study, we combine technical and finance sentiment indicators, this method both considers investors' trading willing and historical stock data and extracts more information affecting stock movement direction.

With the advance in technology, the dimension and quantity of data have greatly improved. Data in the stock market has nonstationary and nonlinearity characteristics, which influence the model's performance. Variable selection technology is an effective approach to dealing with the situation, including the Principal Component Analysis (PCA) (*Wold, Esbensen & Geladi, 1987*), Least Absolute Shrinkage and Selection Operator (LASSO) (*Tibshirani, 2011*). LASSO method is a compression estimation model, which obtains a more refined model by constructing a penalty function that compresses some coefficients. Mathematically, LASSO is a biased estimator that processes data with complex covariance and retains the advantage of subset shrinkage. The method has been applied in numerous fields. For example, *Ma et al. (2018)* adopts the LASSO penalty model to select the predictor variables from the original dataset and predict the oil volatility. The result indicates that the variable selection of LASSO can improve prediction accuracy. *Li, Liang & Ma (2022)* employ the MIDAS-RV framework and construct the MIDAS-LASSO model to

predict the stock price fluctuation based on economic policy and financial stress indicators, the improved model has superior prediction performance under the COVID-19 epidemic.

Existing studies mostly improve the prediction method from a single perspective, such as models, indicators and variable selection. In this article, we combine sentiment analysis and technical indicators to capture more information reflecting stock movement. To avoid the noise and multi-collinearity of predictor variables and process the time-series data, the LASSO-LSTM model is applied to predict the stock movement direction.

The contributions of this article are as follows:

- The existing research only focuses on the technical indicators or finance sentiment indicators. Here we crawl the financial textual content and stock history price from websites, then calculate the sentiment indicators and technical indicators. It includes more information reflecting stock prices and trading willingness compared to the existing indicator system.
- We adopt the FinBERT model to extract textual sentiment information from massive finance data. This transfer learning model is pre-trained by massive finance textual content and has superior performance in finance sentiment analysis.
- To eliminate multi-collinearity and noise between predictor variables, we propose the LASSO-LSTM model, which has the memory structure and can complete variable selection. Finally, we compared the prediction performance of LASSO-LSTM, RF, LSTM, and SVM based on 4 metrics.

The remainder of this article is organized as follows: "Related Works" describes some related studies about stock prediction methods; "Materials & Methods" introduces the architecture of the method proposed, including the data collecting processing, variable selection and indicators calculation; In "Experiment & Results", we train the model and compare the performance of LASSO-LSTM, RF, SVM and LSTM based on 4 evaluation metrics. "Conclusion" summarises the studies and proposes some future research directions.

## RELATED WORKS

This section will describe studies related to LSTM neural networks, stock price forecast, text sentiment analysis, and the BERT model in the following.

### LSTM model

To improve the long-term dependencies problem in the RNN model, a gating mechanism is introduced in the RNN model, which is called gated RNN. The gating mechanism can control the rate of information accumulation, including adding or forgetting the information. The LSTM model is a variant of the RNN model, which with multiple gating mechanisms can effectively solve the gradient disappearance problem in the simple RNN model. The improvements of LSTM to RNN focus on two points: (1) Introduce the internal state to transmit the information; (2) Adopt the gating mechanisms to control the speed of information accumulation. In this article, we combine the LASSO method and LSTM

model, eliminating noise and multi-collinearity in the data by variable selection, and then improve the prediction accuracy of LSTM.

## Finance sentiment analysis

The stock market fluctuated due to various factors influence, such as the economic environment, political events and investors' sentiments. Investors will expect the stock price to rise or fall according to information they could obtain, such as financial news and web comments. Therefore, analysing financial textual content can recognise the sentiment of the market and trading willingness, and then predict the behaviour of investors (*Ko & Chang, 2021*). Prior studies in finance textual content analysis suggest that financial articles and social media comments can be applied to predict stock prices (*Tetlock, Saar-Tsechansky & Macskassy, 2008*). However, the quantitative analysis of textual content is complex due to the noise and unstructured characters of the data (*Renault, 2020*)

In this context, researchers adopt the natural language processing (NLP) methodology to study sentiment analysis problems in the finance field. The most common analysis approach in textual data sentiment content is the dictionary-based approach, machine learning and deep learning. The dictionary-based approach extracts the sentences from textual content to be analyzed and then splits the sentences based on the stop-word problems. Each word spelt from sentences has its sentiment tendency and can be classed into the corresponding categories, based on a pre-defined emotion dictionary (*Li et al., 2010*; *Tulkens et al., 2016*). Therefore, we could distinguish the emotion of textual content according to the word labels. Compared with machine learning and deep learning, the advantage of the dictionary-based approach is that no label of the data is required, and no training is needed.

To achieve higher prediction accuracy, some machine learning methods are applied to the sentiment analysis field, such as Linear Discriminant Analysis (LDA), Support Vector Machines (SVM) and naive Bayes (NB). Machine learning methods can process the noise in the textual data and retain the standard text content. Afterwards, the pre-processed data is subjected to feature extraction, which results in a feature vector to indicate the sentiment indicators of textual content. *Moraes, Valiati & Neto (2013)* compare the ANN and SVM model performance based on the document-level sentiment analysis tasks. The result suggests that the classification accuracy of ANN outperforms SVM significantly; *Kanakaraj & Guddeti (2015)* proposes the improved ensemble NLP method, which considers semantics in word vectors and performs more accuracy than a single classifier. In this article, we collect the daily finance news of stocks and then calculate the sentiment indicators such as positive score, neutral score, negative score and daily article numbers by the FinBERT model.

## FinBERT

With the improvement of computing technology, researchers begin to pre-train models based on large corpora and then use the pre-trained models in NLP tasks, where such models have good performance (*Peters et al., 2018*; *Howard & Ruder, 2018*). The Bi-directional Encoder Representation from Transformers (BERT) model is pre-trained on general

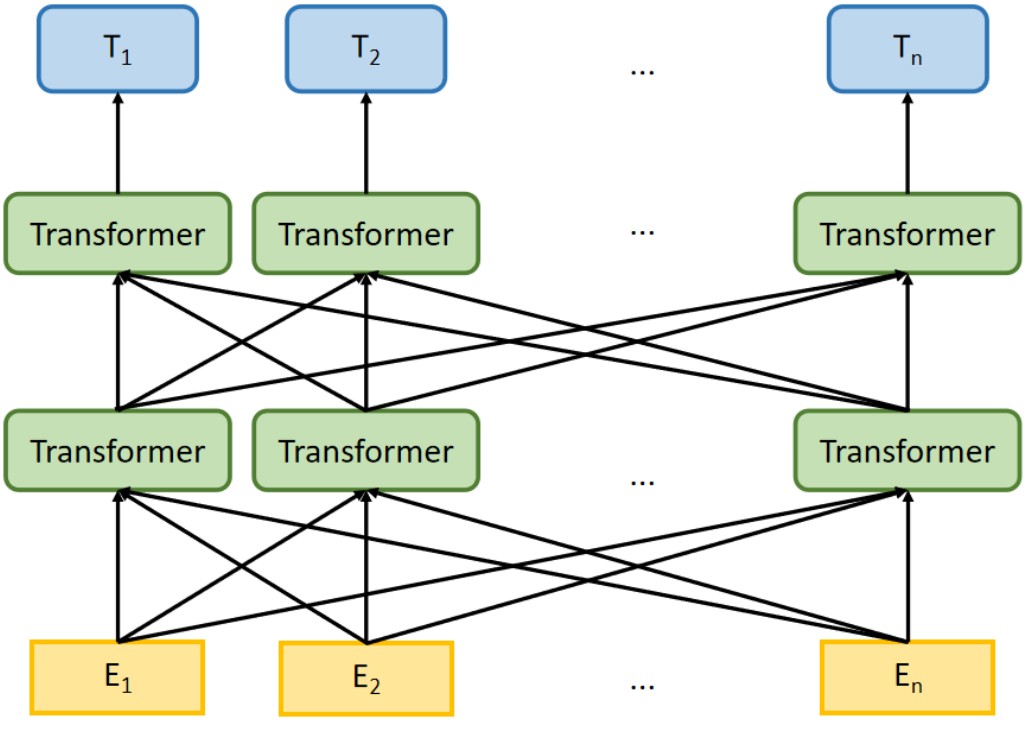

**Figure 1** **The architecture of BERT model.**

domain textual content, such as news articles and Wikipedia (*Devlin et al., 2018*). Figure 1 is the architecture of the BERT model, $E_1, E_2, \ldots, E_n$ represent the inputs, $T_1, T_2, \ldots, T_n$ are outputs of BERT model, which computed by multi-layer transformer encoder. BERT uses the transformer encoder block for the concatenation instead of the traditional Transformer decoder, the former is a typical bi-directional encoder model.

Experiments indicate that BERT has superior classification capabilities to traditional NLP models, improving the accuracy by 5.6% on the MultiNLI dataset (*Devlin et al., 2018*). However, on some specialized NLP tasks, BERT models pre-trained based on general domain textual content cannot process some proprietary vocabularies and performs poorly. To this end, *Liu et al. (2021)* present FinBERT (BERT for Financial Text Mining) which is pre-trained on large-scale financial and business corpora and has enhanced performance. In this study, we adopt the FinBERT, a domain-specific BERT model for finance sentiment analysis tasks, the details of FinBERT architecture are listed in Fig. 2.

FinBERT is proposed based on the standard BERT model, which is a two-stage pre-training model that includes pre-training and fine-tuning. Compared with standard BERT, FinBERT applied the varying pre-training methods from standard BERT, the improved model employs multiple self-supervised pre-training tasks that can be trained by multi-task self-supervised learning methods to capture linguistic knowledge and semantic information in large-scale pre-trained corpora more effectively. In the pre-training phase, the training finance textual data of FinBERT includes four parts: English Wikipedia & BooksCorpus, FinancialWeb, YahooFinance and RedditFinanceQA. In the fine-tuning

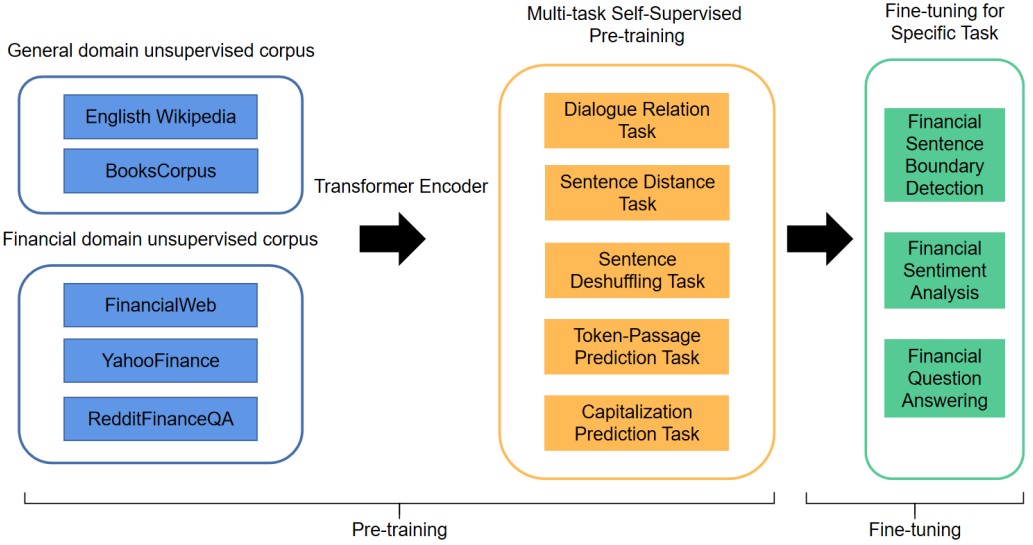

General domain unsupervised corpus

Englisth Wikipedia

BooksCorpus

Financial domain unsupervised corpus

FinancialWeb

YahooFinance

RedditFinanceQA

Transformer Encoder

Multi-task Self-Supervised Pre-training

Dialogue Relation Task

Sentence Distance Task

Sentence Deshuffling Task

Token-Passage Prediction Task

Capitalization Prediction Task

Fine-tuning for Specific Task

Financial Sentence Boundary Detection

Financial Sentiment Analysis

Financial Question Answering

Pre-training

Fine-tuning

**Figure 2** **The illustration of the architecture for FinBERT.**

phase, FinBERT adopts the pre-training parameters and tunes on the specific task. The experiments indicate that FinBERT achieves superior performance than the BERT model on the Financial Phrasebank dataset, 14% higher than SOTA method (*Liu et al., 2021*).

# MATERIALS & METHODS

This section mainly introduces the architecture of the LASSO-LSTM model and the stock data preparation. The LASSO-LSTM model consists of two stages: First, we apply the LASSO algorithm to select the important variables; Then, we use the LASSO model processes the time series data and predict the stock movement direction. In the data preparation phase, we crawl the data from websites and calculate the technical and sentiment indicators by TTR packages.

## LASSO-LSTM model

Most existing studies forecasting stock only consider technical or market sentiment information. In this article, we propose a method that combines technical indicators and financial sentiment analysis. Compared with the standard methodology, the improvement methods include more information relating to trading behaviors.

Existing studies commonly forecast stock only consider technical or market sentiment indicators. To combine the advantages of these methods, we propose a method that adopts both technical and financial sentiment indicators as predictor variables. To the knowledge of the complexity and correlation in financial stock predictor variables, we hope to eliminate the variables unimportant relatively to disease the dimensionality of predictor variables and improve the model performance. The structure of the prediction model is shown in Fig. 3. First, we crawl the finical textual content and stock historical transaction information from websites. Afterward, the textual data is split into sentences, and input to the FinBERT model, obtaining the 4 sentiment indicators. Simultaneously,

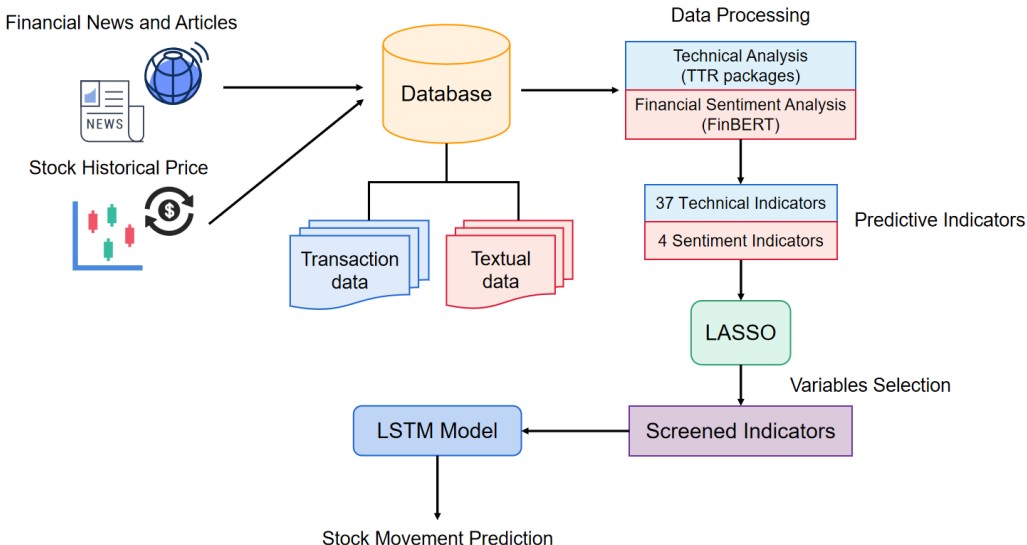

**Figure 3** The architecture of proposed stock prediction model.

37 technical indicators are calculated by TTR packages. In the variable selection phase, we apply the LASSO method to eliminate the variables with high correlation. Finally, we adopt the indicators after variable selection as the input vectors to train the LASSO-LSTM model and then evaluate the model performance on the test data set.

### LASSO algorithm

Lasso logistic regression is a model that incorporates the $L_1$ norm as a penalty function, in contrast to ordinary logistic regression models, which yield a sparse solution. Here we assume that the predictor variables is $X = (x_1, x_2, \ldots, x_n)$, response variables is $Y = (y_1, y_2, \ldots, y_n)$. $c_t$ represents the closing price of a given stock in $t$-th day, $z_{t+1} = c_{t+1} - c_t$ represents the stock return in $t+1$-th day. The stock movement direction function can be expressed as follows

$$y_{t+1} = \begin{cases} 1, z_{t+1} > 0 \\ 0, z_{t+1} \leqslant 0 \end{cases}, t = 1, \ldots, n-1. \tag{1}$$

where $y_{t+1}$ represents the stock prices movement direction in $t+1$-th day. Then, the conditional probabilities of up and down trends for stock prices can be expressed as:

$$P(y_{t+1} = 1|x_t) = \frac{e^{\beta_0 + x_t\beta}}{1 + e^{\beta_0 + x_t\beta}}, \tag{2}$$

generally, the $\beta_0$ is intercept of the predictor variables, $\beta = (\beta_1, \ldots, \beta_p)^\top$ is the parameter vector of predictor variables, and $x_t = (x_{t,1}, \ldots, x_{t,p})$ is the predictor variables in $t$-th day. This function is the sigmoid function and can be used to describe binary-classification datasets. Based on the above probabilities function and data set, we get the log-likelihood function as

$$L(\beta_0, \beta) = \sum_{i=1}^{n} \log\left(P(y_{t+1}|x_t)\right). \tag{3}$$

Compared with standard logistic regression, the LASSO algorithm introduces the penalty function, and the LASSO criterion is

$$Q(\beta_0, \beta; \lambda) = L(\beta_0, \beta) + \lambda P_\alpha(\beta), \tag{4}$$

where $P_\alpha(\beta) = \sum_{j=1}^{p} |\beta_j|$, $|\beta_j|$ is the penalty function, $\lambda \geqslant 0$ is a tuning parameter that controls the intensity of penalization, $p$ is the number of indicator. *Breheny & Huang (2011)* introduced the coordinate descent algorithm to solve the penalized weighted least-squares problem and get the iterative estimation of LASSO.

$$\min_{(\beta_0, \beta) \in \mathbb{R}^{p+1}} \{-L(\beta_0, \beta) + \lambda P_\alpha(\beta)\}. \tag{5}$$

The LASSO algorithm is characterized by the ability of convergence and variable selection. It shrinks the range of parameters to be estimated and only a few parameters are estimated at each step. In addition, the LASSO algorithm automatically selects the critical variables to build the model. In the stock prediction problem, the LASSO algorithm can eliminate the multi-collinearity among predictor indicators and avoid model over-fitting, disease the calculation time of the model. Here we provide the pseudo code on how to apply the LASSO algorithm to obtain estimators.

---

**Algorithm 1:** The solution process of LASSO algorithm

---

**Input**: The training set $\{(x_{t,1}, x_{t,2}, \ldots, x_{t,p}), y_{t+1}\}_{t=1}^{N}$, the optimal $\lambda$, the maximum iteration number $M$ and the tolerance limit $\varepsilon$

1 **Initialization**: Set $m = 0$, $\widetilde{\beta} = (\widetilde{\beta}_0, \widetilde{\beta}_1, \ldots, \widetilde{\beta}_p)^\top$, $\widetilde{\beta}(0)$ be the initial value of the parameter estimated vector, $\widetilde{\beta}(0) \leftarrow 0$,

2 **repeat**

3    **for** $j = 1, 2, \ldots, p$ **do**

4      the training set $X$, where the $x_{\cdot j} = (x_{1,j}, x_{2,j}, \ldots, x_{N,j})^\top$, the $x_{t \cdot} = (x_{t,1}, x_{t,2}, \ldots, x_{t,p})^\top$,

5      $\widehat{P}_t = \frac{e^{\widetilde{\beta}_0(m) + x_{t \cdot} \widetilde{\beta}(m)}}{1 + e^{\widetilde{\beta}_0(m) + x_{t \cdot} \widetilde{\beta}(m)}}$,

6      $\omega_t = \widehat{P}_t(1 - \widehat{P}_t)$, $W = \mathrm{diag}(\omega_1, \omega_2, \ldots, \omega_N)$,

7      $z_j = n^{-1} x_{\cdot j}^\top W x_{\cdot j} + v_j \widetilde{\beta}_j(m)$,

8      $v_j = n^{-1} x_{\cdot j}^\top W x_{\cdot j}$,

9      $\widetilde{\beta}_j(m+1) \leftarrow \frac{S(z_j, \lambda)}{v_j}$

10      where $S(z_j, \lambda) = \mathrm{sign}(z_j)(|z_j| - \lambda)_+ = \begin{cases} z_j - \lambda, z_j > 0, \lambda < |z_j|, \\ z_j + \lambda, z_j < 0, \lambda < |z_j|, \\ 0, \lambda \geq |z_j|. \end{cases}$

11    **end**

12 **until** $\left\| \widetilde{\beta}(m+1) - \widetilde{\beta}(m) \right\|_2^2 \leq \varepsilon$ *or* $m = M$;

**Output**: $\widetilde{\beta}(m+1)$

---

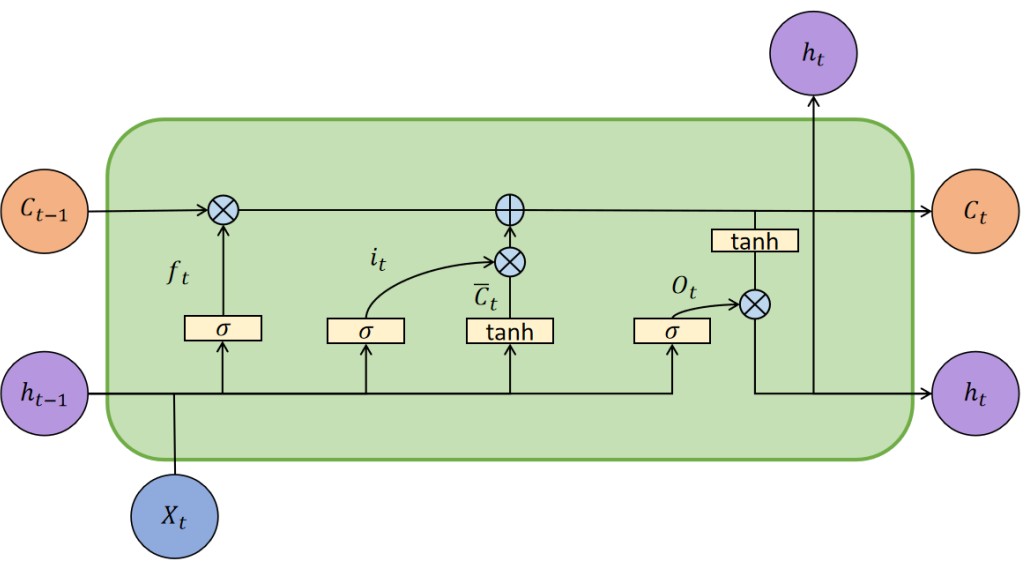

**Figure 4  The architecture of the LSTM model.**

### LSTM model

Based on the above work, we obtain the indicators after variable selection and could predict the stock movement direction. LSTM has the gating structure to control the transmission of information. Figure 4 is the architecture of the LSTM model.

In Fig. 4, $X_t$ is the input vector at $t$ moment, $h_t$ is the external state, and $C_t$ is the internal state, which performs circular messaging and also outputs information to the external state $h_t$ of the hidden layer. The $C_t$ calculated as follows:

$$C_t = f_t \odot C_{t-1} + i_t \odot \overline{C}_t, \tag{6}$$

$$h_t = O_t \odot tanh(C_t), \tag{7}$$

where $C_{t-1}$ represents the memory unit in $t-1$ moment, $\overline{C}_t$ is the candidate state calculated by no-linear function:

$$\overline{C}_t = tanh(W_c X_t + U_c h_{t-1} + b_c), \tag{8}$$

where $W_c$ and $U_c$ represent the weight matrices, and $b_c$ represents the bias values vector. Tanh function is a activation function. LSTM control the path of message delivery by gating mechanism, the function as follows:

$$i_t = \sigma(W_i X_t + U_i h_{t-1} + b_i), \tag{9}$$

$$f_t = \sigma(W_f X_t + U_f h_{t-1} + b_f), \tag{10}$$

$$O_t = \sigma(W_O X_t + U_O h_{t-1} + b_O) \tag{11}$$

where $\sigma$ is the sigmoid function, which output is 0 to 1. Forget gate $f_t$ controls the information needs to be forgotten from last internal state $C_{t-1}$; input gate $i_t$ controls the information needs to be input from candidate state $\overline{C}_t$. At moment $t$, the mechanism of the lstm model as follows: (1) Calculating the $i_t, f_t$ and $o_t$ based on last external state $h_{t-1}$ and input vector; (2) Combining $f_t$ and $i_t$ to update memory units $C_t$; (3) Passing information from the internal state to the external state $h_t$.

### LASSO and LSTM integrated forecasting algorithm

We combine the LASSO and LSTM models for short-term stock movement forecasting. Set $D = \{X_t, Y_t\}_{t=1}^{N}$ represent the stock data set. The LASSO-LSTM can be divided into stages: first, we take the stock data set to the LASSO algorithm to select the important variables and eliminate the multi-collinearity among predictor variables. Then, we adopt the grid search and 10-fold cross-validation to tune the parameter value in LASSO model; Finally, we apply the predictor variables after selection as the input vector into LSTM model and predict the stock movement direction. We summarize the procedure of the LASSO-LSTM model in Algorithm2, which can execute variables selection and effectively process the time-series data set due to the penalized function and gate structure.

---

**Algorithm 2:** The algorithm produce of LASSO-LSTM model

---

1  **Initialization** : The predictor variables $\{X_t\}_{t=1}^{N}$ are normalized to eliminate scale effects.

2  Divide the data set $D = \{X_t, Y_t\}_{t=1}^{N}$ to the training set and testing set;

3  Adopt the LASSO algorithm to variables selection, eliminate the multi-collinearity among predictor variables;

4  Set the alternative parameter values $\{\lambda_1, \ldots, \lambda_K\}$, where $0 \leq \lambda_k \leq 1. \lambda_k < \lambda_{k+1}, \mathrm{k}=1, \ldots, K$. Tuning the $\lambda$ parameter values in the LASSO algorithm by grid search and 10-fold cross-validation method based on training set;

5  Select the $\lambda$ value with the best performance and retain the corresponding critical predictor variables in the training set and testing set;

6  Predict the stock movement direction based on the LSTM model and variables after processing;

---

Note that to make sure the parameter tuning is effective, we rank the alternative parameters from smallest to largest to get $\{\lambda_1, \ldots, \lambda_K\}$. The selection result of each lambda value was evaluated by grid search and 10-fold cross-validation. Too small lambda values will weaken the variable selection ability of the lasso algorithm, and too large lambda values will lead to the loss of important information in the data. Such a parameter tuning method could guarantee reasonable results and efficient computation. In this way, we construct the

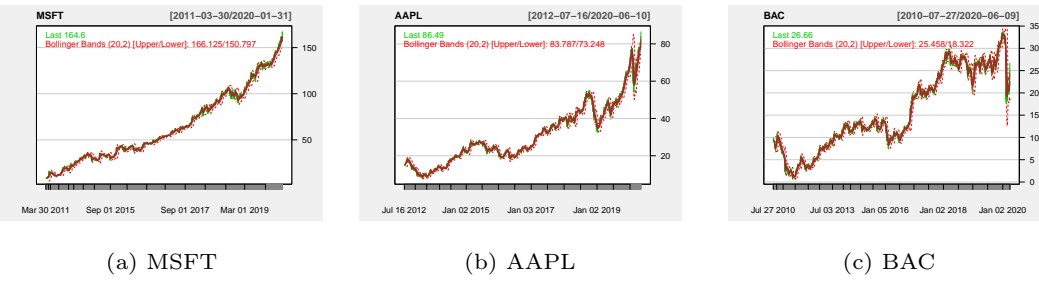

(a) MSFT        (b) AAPL        (c) BAC

**Figure 5   The historical stock prices for MSFT (A), AAPL (B) and BAC (C).**

LASSO-LSTM model that could reduce the dimensionality of the predictor variables and capture key information in time-series data.

Compared with traditional machine learning models, the proposed LASSO-LSTM has advantages when applied to stock prediction problems: (1) The technical indicators and sentiment indicators in stock prediction always have multi-collinearity, different from standard LSTM, LASSO-LSTM could eliminate these factors on prediction performance; (2) In the prediction phase, the LASSO-LSTM model predicts the time-series stock data by the LSTM model rather standard LASSO model, which can capture the time-series more effectively; (3) The LASSO-LSTM model has fewer hyperparameters, so it does not require several numbers of computational resources to achieve good prediction accuracy.

## Data preparation
### Data crawling

In this article, we hope to fuse the finical textual data and stock historical transaction data and forecast stock movement more accurately. To this goal, we need to select some companies and collect their transaction and financial news data. First, we select three representative stocks as analysis subjects from NYSE/NASDAQ: AAPL, MNST, and BAC.

In stock history data collection phase, we apply the E-finance packages in Python to crawl three companies' stock data derived from 2012 to 2020. Stock historical transaction data set includes Opening Price, Closing Prices, Highest Price, Lowest Price, and Trading Volume. Figure 5 plots the three companies' stock price trends, which cover the various situations of stock movement.

Except for historical stock data, the movement of stock prices is also influenced by the trading willingness of investors. So it is significant to grasp the relevant information about stock market emotions. In this study, the textual data is crawled from the investing.com website, which is an online website that provides financial information. The financial textual content consisted of historical news and relevant articles on the US equities' publicly traded information. The history data starts from 2012 to 2020, for different companies, the data size varies. Table 1 is the example of textual data after cleaning, which consists of five indicators.

| Table 1 | The example of financial textual data. | | | |
| --- | --- | --- | --- | --- |
| Article_Id | Ticker | Title | Content | Date |
| 2060327 | NIO | What's happening? | Shares of Chinese electric car maker NIO were ... | 2020-1-15 |

**Notes.**
Article_Id: The number the article was collected.
Ticker: The ticker symbol, represents uniquely a particular stock.
Title: The title of articles.
Content: The details of news.
Date: The date of news released.

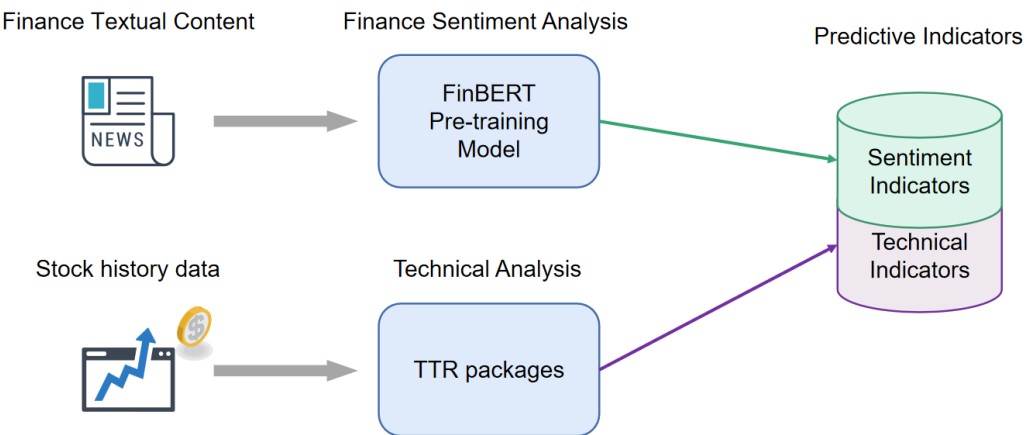

**Figure 6** The processing flow of raw data.

### Technical and sentiment indicators

In the data crawling stage, we collect the stock historical transaction data and financial news data from websites. The raw data is characterized as unstructured and has an amount of noise. To transform the raw data collected above into the predictor indicators, we need to use technical analysis and sentiment analysis technology to calculate the predictor indicators. In this section, we apply TTR package to calculate the 37 technical indicators based on 6 stock historical transaction indicators and obtain 4 sentiment indicators by the FinBERT model. The details of the data processing flow are shown in Fig. 6.

In Fig. 6, we can find that the data crawled from websites consists of historical and textual data. To access information that affects stock fluctuation, it is necessary to process the raw data and obtain predictor indicators. The process includes two stages, one stage is to calculate the sentiment scores from the textual content, and another stage is to calculate the indicators that reflect the changes patterns from the past stock prices.

Regarding technical indicators, scholars proposed that technical analysis could find the relationship between stock movement direction and capture the pattern of stock movement (*Tsai & Hsiao, 2010*). *Murphy (1999)* systematically studied the technical analysis of the stock market and proposed some technical indicators calculated formulas from the stock past prices. Recently, *Yang, Hu & Jiang (2022)* proposed group SCAD/MCP penalized logistic regression to predict up trends and down trends for stock prices with 24 technical

**Table 2  Details of technical indicators.**

| Technical indicators | Description |
| --- | --- |
| ADX | Average Directional Movement Index |
| Aroon oscillator | Aroon oscillator |
| AroonUp | Aroon up Index |
| AroonDn | Aroon down Index |
| ATR | Average True Range Index |
| BBands | Bollinger Bands Index |
| CCI | Commodity Channel Index |
| CMF | Chaikin Money Flow index |
| CMO | Chande Momentum Oscillator |
| DC | Donchian Channels Index |
| DPO | Detrended Price Oscillator |
| DVI | David Varadi Intermediate Index |
| GMMA | Guppy Multiple Moving Average Index |
| KST | Know Sure Thing Index |
| MACD | Moving Average Convergence Divergence Index |
| RSI | Relative Strength Index is an momentum oscillator |
| SAR | Stop-and-Reverse Index |
| WMA | Weighted Moving Average Index |
| DEMA | Double Exponential Moving Average Index |
| SMA | Simple Moving Average Index |
| EMA | Exponential Moving Average Index |
| TDI | Trend Detection Index |
| VHF | Vertical Horizontal Filter Index |
| EVWMA | Elastic, Volume-Weighted Moving Average Index |
| ZLEMA | Zero Lag Exponential Moving Average Index |
| WAD | Williams Accumulation / Distribution Index |
| HMA | Hull Moving Average Index |
| SMI | Stochastic Momentum Index |
| BR | BR indicator reflects the degree of willingness to trading |
| AR | AR indicator reflects the market's buying and selling sentiment |
| OBV | On Balance Volume Index |
| Chaikin AD | Chaikin Accumulation / Distribution (AD) line |
| CLV | Close Location Value Index |
| MFI | The Money Flow Index |
| OBV | On Balance Volume Index |
| ROC | Rate of Change Index |

indicators. To this end, we calculate 37 commonly used technical indicators by the Technical Trading Rules(TTR) package. The details are shown in Table 2.

TTR is a package in R software, which could calculate technical indicators according to technical trading rules based on stock historical transaction data set, details of the calculation formulas see the URL: https://github.com/joshuaulrich/TTR. The indicators that

can be calculated by TTR include moving average trend indicators, oscillators, volatility indicators, and volume indicators.

After obtaining the 37 technical indicators, we hope to calculate sentiment indicators from textual content data. *Baker & Stein (2004)* started with market liquidity and established a model to explore the internal mechanisms of sentiment indicators and stock price fluctuations. In this article, raw textual data consists of a vast amount of articles and news. We extract the news of target companies and then split the content into sentences. Sentiment analysis aims to evaluate textual content based on sentences and calculate the scores for each emotion. The sentiment score is a continuous numeric value and applied to obtain sentiment probabilities for articles as follows:

$$
\begin{aligned}
P_{pos} &= \frac{N_{pos}}{N_{total}}, \\
P_{neg} &= \frac{N_{neg}}{N_{total}}, \\
P_{neu} &= \frac{N_{neu}}{N_{total}}, \\
P_{pos} + P_{neg} + P_{neu} &= 1,
\end{aligned}
\tag{12}
$$

where $P_{pos}$ is the probability of the article with positive sentiment. $N_{pos}$ denotes the number of sentences with positive sentiment, and $N_t$ denotes the number of total sentences. Similarly, $P_{neg}$ and $P_{neu}$ represent the probability of the article with negative and neutral sentiments. In this article, we calculate the positive/negative/neutral emotional probabilities as the sentiment indicators.

In the sentiment analysis phase, first, we extract the textual content related to the target company, arranging them by date. Afterward, the daily text content is cleaned to remove punctuation and split into sentences. Finally, we input the sentences to the FinBERT model and obtain four sentiment indicators, namely Articles_Number, $P_{pos}$, $P_{neg}$ and $P_{neu}$. The example of sentiment indicators after being processed is listed in Table 3, where the Articles_Number denotes the number of daily article news. After the above pre-processing of data, we transform the unstructured raw data into the 41 predictor variables.

### Exploratory data analysis

Based on the fusion of 37 technical indicators and 4 sentiment indicators, we obtain the 41 predictor variables to forecasting. We perform an exploratory analysis of the predictor variables, calculate the correlation coefficients of the predictor variables, and visualize them. The details are shown in Fig. 7.

From Fig. 7, we can conclude that the most predictor variables have significant correlation, such as $X_{25}, X_{17}$, and $X_{32}$ etc. The precise correlations or highly correlated relationships among predictor variables can produce multi-collinearity, which influences model stability or make the model parameters difficult to be estimated accurately. In this situation, the LASSO algorithm could eliminate the multi-collinearity by introducing the penalized function to compress some unimportant variables to 0. Due to this advantage,

**Table 3** The example of AAPL sentiment indicators.

| Date | Articles_Number | Positive | Neutral | Negative |
|---|---|---|---|---|
| 2012-7-16 | 1 | 0.8739 | 0.0847 | 0.0412 |
| 2012-7-19 | 1 | 0.8801 | 0.0240 | 0.0958 |
| 2012-7-23 | 3 | 0.5425 | 0.0330 | 0.4244 |

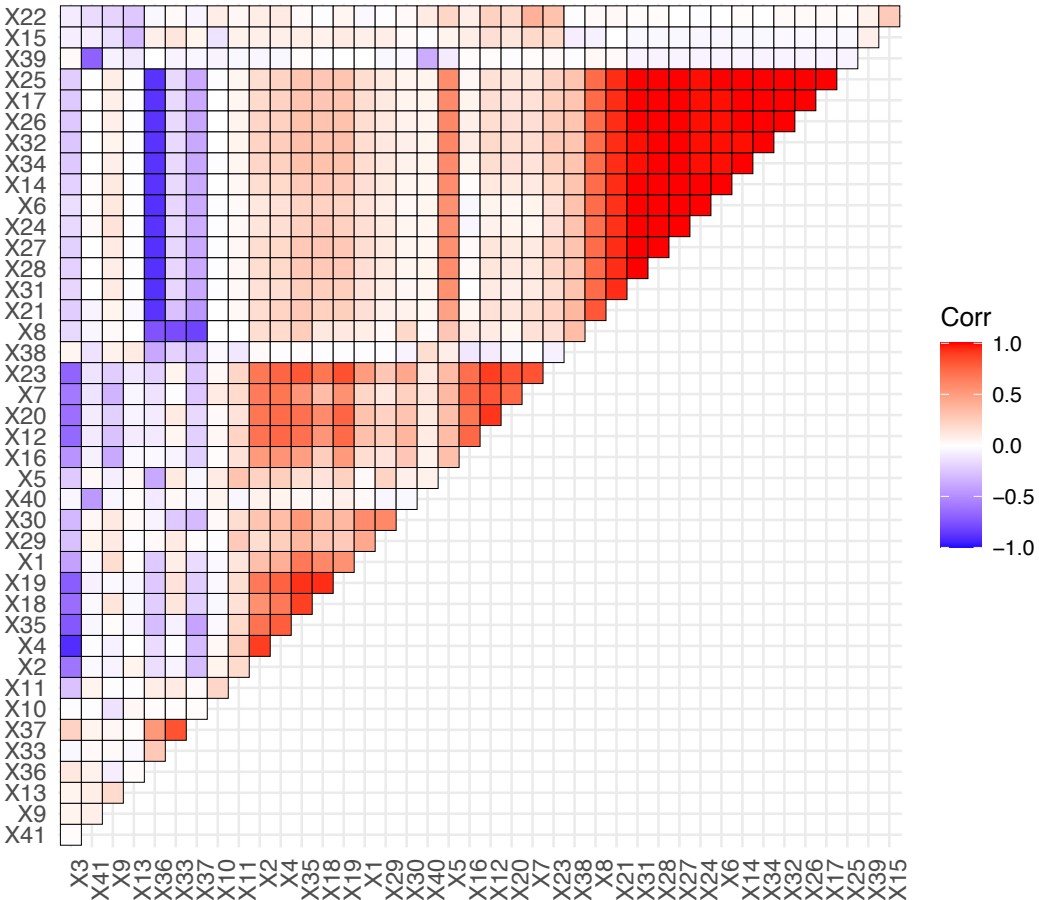

**Figure 7** The correlation of predictor variables.

LASSO-LSTM is an effective method to process the stock data set that combines technical and sentiment indicators.

# EXPERIMENT AND RESULTS

Existing studies only use technical indicators or sentiment indicators as the input vector to train the machine learning models. To achieve superior prediction accuracy, we combine the technical and sentiment indicators to gain predictor variables that cover more long-term information. In this section, the improved LASSO-LSTM model with variable selection is

**Table 4  Confusion matrix.**

|  | Prediction positive | Prediction negative |
|---|---|---|
| Reference positive | Truth positive ($V_{(1,1)}$) | False negative ($V_{(1,2)}$) |
| Reference negative | False positive ($V_{(2,1)}$) | Truth negative ($V_{(2,2)}$) |

used to forecast stock direction. In addition, we compared the prediction performance of LASSO-LSTM with the standard LSTM and classical machine models.

## Evaluation metrics

Stock movement prediction is a binary classification problem, unbalanced sample distribution may cause it difficult for the accuracy metric to realistically assess the predictor performance of the model. Here we introduce the Kappa coefficient, ROC curve, sensitivity, and specificity confusion matrix to evaluate the model prediction performance. *Wan et al. (2015)* proposed the Kappa coefficient to measure the reliability of the model, and the calculated result of the Kappa coefficient was between $[-1,1]$. In $K$ The formula of Kappa is

$$Kappa = \frac{p_0 - p_e}{1 - p_e}, \tag{13}$$

where $p_0$ is the overall prediction accuracy, $p_e = \sum_{i=1}^{K} a_i \times b_i$, $n$ is the number of samples, $a_i$ is the number of $i$ class samples in reference, $b_i$ is the number of $i$ class samples in prediction, $i = 1, \dots, K$. The confusion matrix is a method to visually evaluate the performance of supervised learning algorithms. In Table 4, the $V_{(k,.)}$ represents the number of samples truly belong to $k$ class, and the $V_{(.,k)}$ represents the number of samples be predicted to $k$ class. $V_{(i,j)}$ represents the number of samples belonging to the $i$ class and predicted to $j$ class.

According to confusion matrix, we can calculate the model prediction performance evaluation metrics, such as accuracy, sensitivity and specificity. The accuracy metric represents the ratio of the number of samples with correct model predictions to the number of all samples,

$$Accuracy = \frac{TP + TN}{TP + TN + FP + FN}. \tag{14}$$

The sensitivity metric indicates the number of positive cases correctly predicted as a percentage of the truth number of positive samples,

$$Sensitivity = \frac{TP}{TP + FN}. \tag{15}$$

The specificity metric indicates the percentage of negative cases correctly predicted out of the actual negative samples,

$$Specificity = \frac{TN}{TN + FP}. \tag{16}$$

The above metrics assess the forecasting results from single perspectives but can not reflect the prediction of the model's forecasting performance comprehensively. To this end,

we introduce ROC curves based on specificity and sensitivity metrics. The specificity metric is the $x$-axis, which represents the false positive rate (FDR) and the sensitivity metric is the $y$-axis, which represents the true positive rate (TPR). The lower the FPR metric, the lower the false positive rate of the model and the lower the probability of predicting samples with actual labels of negative as positive; the larger the TPR metrics, the higher the proportion of samples with actual labels of positive being correctly identified in the prediction results. The area under the ROC curve is the AUC metric, which plots the curve based on FPR and TPR metrics at different thresholds. Generally, the larger AUC values indicate better model performance.

## Variable selection

In LASSO-LSTM model, to eliminate the scale effect of the different variables, we make standardization for the predictor variables. Then, we adopt the coordinate algorithm to select the variables. We apply the LASSO algorithm to three companies' data sets and obtain their coefficient path diagrams. The details are shown in Fig. 8, where the $x$-axis is the value of hyperparameter $\lambda$ and the $y$-axis is the value of each coefficient when the iterative algorithm is completed.

Since the value of $\lambda$ will affect the penalized intensity of coefficients and the performance of the model, so we need to choose the optimal value of hyper parameter $\lambda$. By tuning the parameter $\lambda$, we can screen the important variables effectively. Here we use the LASSO algorithm to select the important variables from 41 indicators, with 10-fold cross-validation and grid search to determine the optimal $\lambda$ values. Based on the training set, we choose the optimal $\lambda$ and adopt the LASSO algorithm to select the variables. Then we use the training set data after variable selection and train the LASSO-LSTM model to forecast stock movement. Table 5 lists the variable selection result of three companies.

## Model performance

In this section, we forecast stock movement by the LASSO-LSTM model. To evaluate the prediction method proposed, we analyze the model performance in accuracy metrics, stability, and statistic test. The classic machine learning methods such as ANN, SVM, and RF are introduced in this section. Moreover, we compare the proposed method with the standard LSTM model only with technical indicators (LSTMTI) or the standard LSTM model only with sentiment indicators (LSTMSI) to recognize the impact of sentiment and technical indicators.

Since the financial news release frequency of different stocks varies, we select the period with a high frequency of financial news releases for analysis from three data sets (1000 trading days in total). Considering stock price is characterized by the time series, we adopt the rolling prediction method to evaluate model performance rather than cross-validation. A typical split approach is to split 75% data set to construct the train set and 25% to construct the test set. Here we use the 750 samples from the train set to train the model, and then evaluate the model performance on the 250 samples from the test set. The details are listed in Table 6.

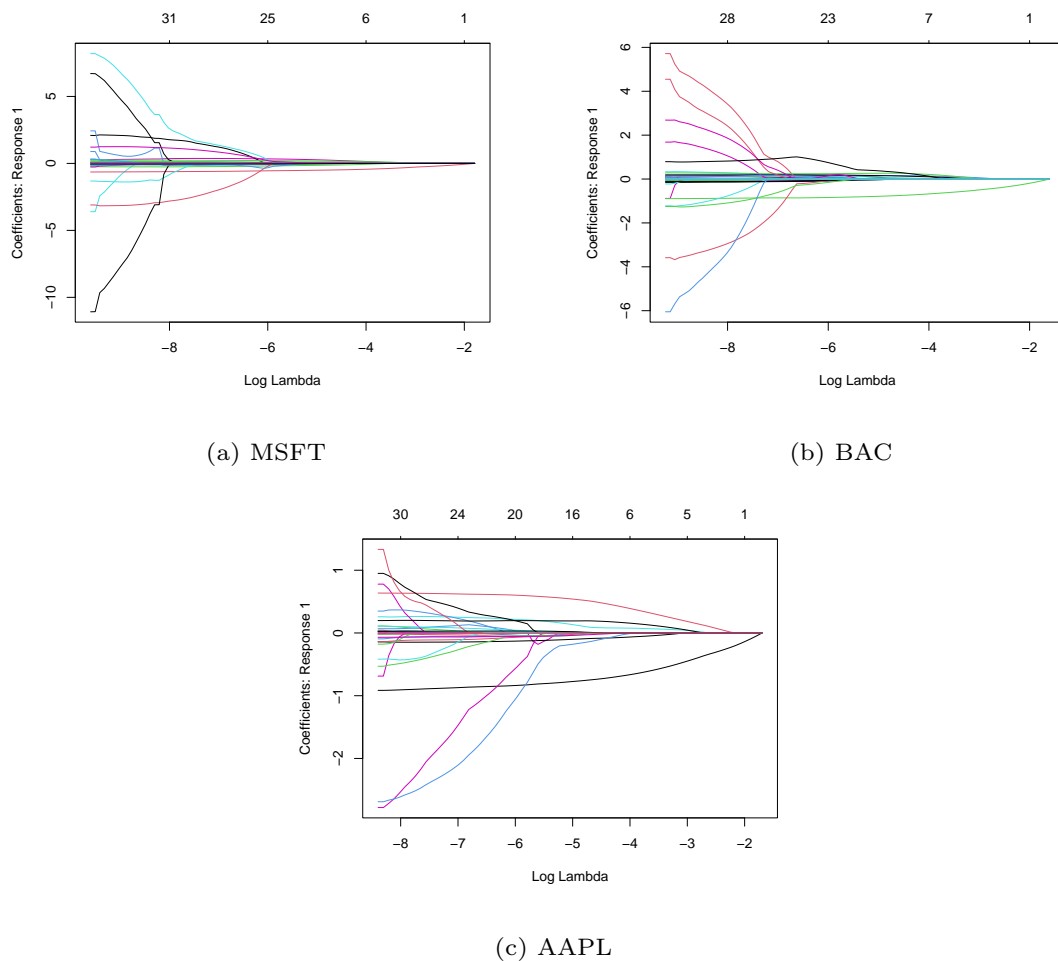

(a) MSFT

(b) BAC

(c) AAPL

**Figure 8** **The coefficient path diagrams based on MSFT (A), BAC (B) and AAPL (C).**

### Model accuracy

To compare the prediction accuracy of the six models, we train LASSO-LSTM, LSTM, LSTMTI, LSTMSI, RF, and SVM based on three companies' stock data. For consistency, the deep learning models use the Adam optimizer for the optimizer and the binary cross-entropy function for the loss function.

Since to the stochastic characteristics of deep learning models, the train models are different in each training phrase. We design the experiment: First, we train each model 30 times based on the stock data set; Then we obtain 30 trained models and evaluate the prediction accuracy on the test set; Finally, we select the trained model with the highest accuracy metrics as the prediction result. Such an experimental design avoids the interference of randomness in the model and makes the experimental results more reliable.

In the evaluation phase, we compare the prediction performance by five evaluation metrics. The prediction accuracy results are listed in Table 7.

In Table 7, it can be seen that the prediction accuracy of the proposed method has an average improvement of 8.533% compared with standard LSTM. We can notice that the

**Table 5  The variable selection result on three data sets.**

| Company | The number of excluded variables |
|---------|----------------------------------|
| MSFT | 14 |
| AAPL | 10 |
| BAC | 16 |

**Table 6  The training sets and the testing sets from MSFT, AAPL and BAC.**

| Company | The training set | The testing set |
|---------|------------------|-----------------|
| MSFT | 2015/12/21-2018/12/02 | 2018/12/21-2019/12/31 |
| AAPL | 2015/12/08-2018/12/06 | 2018/12/07-2019/12/04 |
| BAC | 2014/11/18-2018/03/02 | 2018/03/05-2019/03/13 |

**Table 7  The prediction performance of 6 models.**

| Company | Model | Kappa | AUC | Sensitivity | Specificity | Accuracy |
|---------|-------|-------|-----|-------------|-------------|----------|
| MSFT | LASSO-LSTM | 0.3988 | 0.7347 | 0.6122 | 0.7829 | 0.7160 |
|  | LSTM | 0.2280 | 0.6820 | 0.4796 | 0.7434 | 0.6400 |
|  | SVM | 0.1896 | 0.6269 | 0.4694 | 0.7171 | 0.6100 |
|  | RF | 0.2119 | 0.6524 | 0.7551 | 0.5395 | 0.6240 |
|  | LSTMTI | 0.2068 | 0.5851 | 0.1020 | 0.9211 | 0.6000 |
|  | LSTMSI | −0.0194 | 0.5243 | 0.9737 | 0.0102 | 0.5960 |
| AAPL | LASSO-LSTM | 0.4768 | 0.8119 | 0.5673 | 0.8904 | 0.7560 |
|  | LSTM | 0.3279 | 0.7104 | 0.3942 | 0.9110 | 0.6960 |
|  | SVM | 0.1081 | 0.6323 | 0.2788 | 0.8219 | 0.6060 |
|  | RF | 0.3555 | 0.7066 | 0.7308 | 0.6370 | 0.6760 |
|  | LSTMTI | 0.2726 | 0.6938 | 0.3558 | 0.8973 | 0.6720 |
|  | LSTMSI | −0.0047 | 0.5434 | 0.0096 | 0.9863 | 0.5900 |
| BAC | LASSO-LSTM | 0.5153 | 0.8126 | 0.6154 | 0.8836 | 0.7720 |
|  | LSTM | 0.2340 | 0.6673 | 0.3558 | 0.8630 | 0.6520 |
|  | SVM | 0.1163 | 0.5929 | 0.3365 | 0.7740 | 0.6000 |
|  | RF | 0.3378 | 0.7101 | 0.7115 | 0.6370 | 0.6680 |
|  | LSTMTI | 0.2064 | 0.6362 | 0.2115 | 0.9726 | 0.6560 |
|  | LSTMSI | 0.0369 | 0.5347 | 0.0344 | 1.0000 | 0.5940 |

proposed LASSO-LSTM with technical indicators and sentiment indicators is superior to LSTMTI or LSMSI, it proves that the combination of technical indicators and sentiment analysis is an effective method to improve prediction accuracy. According to FPR and TPR metrics, we visualised the forecast results, as shown in Fig. 9.

From Fig. 9, the LASSO-LSTM model has the larger Area Under Curve, which indicates the model performance outperforms the baselines. Based on the prediction results of three companies verified that the improved LASSO-LSTM outperforms the classical machine learning model such as LSTM, RF, and SVM. Compared with LSTMSI and LSTMTI, the proposed LASSO-LSTM has superior performance in terms of ROC curves or evaluation

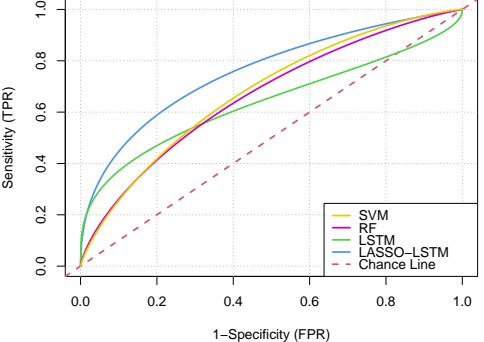

(a) ROC curves of four methods based on MSFT

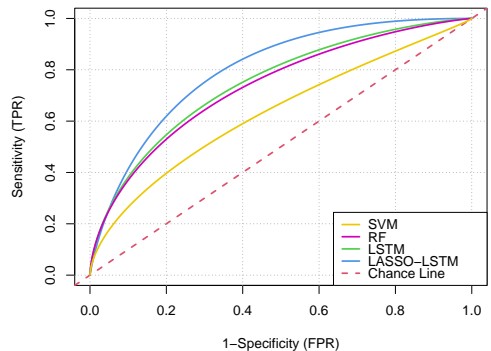

(b) ROC curves of four methods based on AAPL

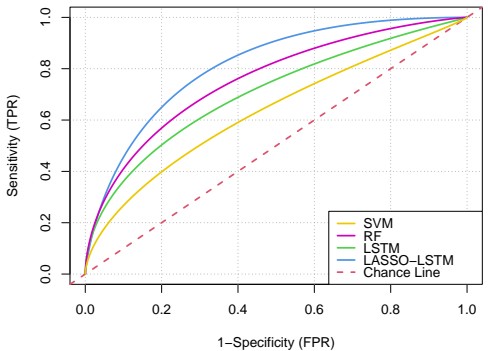

(c) ROC curves of four methods based on BAC

**Figure 9** **The ROC curves based on MSFT (A), BAC (B) and AAPL (C).**

metrics. In addition, the LASSO-LSTM has better performance than the standard LSTM model, it can be concluded that the proposed model can grasp more key information by the fusion indicators and variable selection.

### Model robustness evaluation

In addition to prediction accuracy, robustness is also an important metric in model evaluation. The model with better robustness has better prediction performance despite outliers, missing data, and white noise interference. The model with poor robustness has great fluctuations in prediction results under the interference of random factors, and their applicability is poor and difficult to generalize.

To evaluate the robustness of the model and eliminate the influence of random factors, we repeated the experiment 30 times for the six models and calculated the best accuracy, mean accuracy, and standard deviation of the accuracy. The model with a high standard deviation value indicates that the model is not stable and has poor robustness. The results are shown in the following Table 8.

**Table 8  The prediction accuracy metrics of 6 models.**

| Data set | Method | Best accuracy | Mean accuracy | Sd of accuracy |
|---|---|---|---|---|
| MSFT | LASSO-LSTM | 0.7160 | 0.6806 | 0.0097 |
| | LSTM | 0.6400 | 0.6008 | 0.0285 |
| | LSTMTI | 0.6000 | 0.5586 | 0.3500 |
| | LSTMSI | 0.5960 | 0.5680 | 0.0031 |
| | RF | 0.6240 | 0.5365 | 0.0476 |
| | SVM | 0.6100 | 0.5945 | 0.0047 |
| AAPL | LASSO-LSTM | 0.7560 | 0.7362 | 0.0115 |
| | LSTM | 0.6960 | 0.6240 | 0.0376 |
| | LSTMTI | 0.6720 | 0.5986 | 0.0383 |
| | LSTMSI | 0.5900 | 0.5861 | 0.0008 |
| | RF | 0.6760 | 0.6108 | 0.0410 |
| | SVM | 0.6060 | 0.5896 | 0.0043 |
| BAC | LASSO-LSTM | 0.7720 | 0.7348 | 0.0130 |
| | LSTM | 0.6520 | 0.6060 | 0.0276 |
| | LSTMTI | 0.6560 | 0.5840 | 0.0383 |
| | LSTMSI | 0.5940 | 0.5761 | 0.0007 |
| | RF | 0.6680 | 0.5417 | 0.0705 |
| | SVM | 0.6000 | 0.5896 | 0.0038 |

From Table 8, we can conclude that: (1) The proposed LASSO-LSTM model outperforms other models in terms of Best Accuracy, Mean Accuracy, Sd of Accuracy, etc. Therefore, it can be concluded that the LASSO-LSTM model proposed in this article is more effective than the baseline model. (2) Traditional machine learning methods such as SVM and RF are less effective than LSTM and LASSO-LSTM models in terms of Mean Accuracy and SD accuracy, and it can be seen that for the multivariate time series prediction task in this article, using recurrent neural networks in the form of LSTM is more effective than traditional machine learning models.

### Model statistics test

The Wilcoxon signed rank test determines whether there is a significant difference between two models by constructing a statistic for the difference between the predicted results of different models. In this article, we use the Wilcoxon signed rank test to compare whether there is a statistical difference between the proposed LASSO-LSTM model and baselines. We set the null hypothesis: the prediction results of two models have no statistical differences; the alternative hypothesis: the prediction results of two models has statistical differences.

Table 9 reports statistical test results for the proposed LASSO-LSTM model and five baselines. If $P$-value $\leq 0.05$, the two models have a statistical difference, the positive statistic difference value indicates the proposed model outperforms the baselines. The statistical test results indicate that the performance of LASSO-LSTM model is superior to all other 5 models.

**Table 9  Wilcoxon signed rank test of 6 models.**

| Model 1 | Model 2 | Difference | 95% CI | P-value | Reject |
|---|---|---|---|---|---|
| LASSO-LSTM | LSTM | 0.0740 | (0.0680,0.0820) | 1.790e−06 | True |
| | LSTMTI | 0.0900 | (0.0760,0.0979) | 2.112e−06 | True |
| | LSTMSI | 0.0859 | (0.0819,0.0900) | 1.705e−06 | True |
| | SVM | 0.1459 | (0.1240,0.1640) | 1.823e−06 | True |
| | RF | 0.0680 | (0.0640,0.0720) | 1.809e−06 | True |

## CONCLUSION

In recent years, machine learning and deep learning technology have been applied widely to forecast the stock price. Scholars hope to establish models with excellent predictor performance to predict the trend of stocks more accurately and help investors obtain considerable returns. However, since the nonlinear complex characters of financial data, it is not easy to achieve excellent prediction accuracy by machine learning or deep learning methods. According to previous studies, researchers usually only use one method to construct the prediction model. The technical analysis, NLP, LASSO and LSTM are used in the proposed prediction methodology in this article. Compared with the prediction model that only adopts technical indicators or market sentiment data, we capture more information and predict the stock movement more precisely.

The methodology proposed in this article utilizes sentiment analysis and technical indicators to construct predictor variables and improves the standard LSTM model with the LASSO algorithm to forecast stock movement. Compared with standard models, the LASSO-LSTM has the ability to select the critical predictor variables, it enables the model will not to be affected by multi-collinearity and has better performance. The experiment result demonstrates that the LASSO-LSTM model with combination indicators has superior prediction accuracy significantly than the standard LSTM model. In addition, compared with LSTMSI and LSTMTI, the LASSO-LSTM with a combination of indicators has an improvement of 15.46% and 10.53% respectively. It can be concluded that the sentiment analysis and technical analysis all include critical information about trading signals, and the combined indicators with a variable selection model can improve the prediction accuracy.

There are a few shortcomings. First, we only researched the stock from the NYSE/NASDAQ. Therefore, we are unable to assess the performance of the proposed approach in the equity markets of other countries. In the follow-up research, we will try to apply the proposed model to study the stock market in other countries. Second, model optimization is the method to improve and achieve superior performance in machine learning. It would be worth using the genetic algorithm or particle swarm optimization to tune the hyperparameters and improve the model prediction accuracy.

### Funding

This article is supported by the Graduate Innovation Research Project of Chongqing Technology and Business University, China (yjscxx2022-112-06). The funders had no role in study design, data collection and analysis, decision to publish, or preparation of the manuscript.

### Grant Disclosures

The following grant information was disclosed by the authors:
The Graduate Innovation Research Project of Chongqing Technology and Business University, China: yjscxx2022-112-06.

### Competing Interests

The authors declare there are no competing interests.

### Author Contributions

- Junwen Yang conceived and designed the experiments, performed the experiments, performed the computation work, prepared figures and/or tables, and approved the final draft.
- Yunmin Wang conceived and designed the experiments, performed the experiments, analyzed the data, prepared figures and/or tables, authored or reviewed drafts of the article, and approved the final draft.
- Xiang Li conceived and designed the experiments, analyzed the data, prepared figures and/or tables, authored or reviewed drafts of the article, and approved the final draft.

### Data Availability

The data and codes are available in the Supplemental Files.

Some raw data of financial textual content are available at Kaggle: https://www.kaggle.com/datasets/miguelaenlle/massive-stock-news-analysis-db-for-nlpbacktests.

### Supplemental Information

Supplemental information for this article can be found online at http://dx.doi.org/10.7717/peerj-cs.1148#supplemental-information.

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
