# Peer review of "Prediction of stock price direction using the LASSO-LSTM model combines technical indicators and financial sentiment analysis"

_PeerJ Computer Science, doi:10.7717/peerj-cs.1148_

## Round 0.1 · original submission · Major Revisions

I think this paper needs to be revised to meet the standards of the journal before it is published. Please make reasonable modifications according to the comments of the reviewers, and I look forward to seeing the revised version!

Reviewer 1 ·

Basic reporting

Overall, the quality of this article is very good.Clear and unambiguous, professional English used throughout.Literature references, sufficient field background/context provided.Professional article structure, figures, tables. Raw data shared.

Experimental design

Original primary research within Aims and Scope of the journal.Research question well defined, relevant & meaningful. It is stated how research fills an identified knowledge gap.Methods described with sufficient detail & information to replicate.

Validity of the findings

All underlying data have been provided; they are robust, statistically sound, & controlled.Conclusions are well stated, linked to original research question & limited to supporting results.

Additional comments

1.This paper compares with several common prediction methods, but is there any related research that also uses LASSO-LSTM?
If yes, it needs to be compared. If no, it needs to be explained.
2.The formula should be centered.
3.There is too much white space in some places, as shown before and after Table 3.

Reviewer 2 ·

Basic reporting

(1) The language presentation of this paper is fine, but some sentences can be re-written to show a clear meanings. The main content need to be revised is mostly in section "materials and methods";
(2) The literatures in this paper are mostly old, thus the motivations are weaken in "Introduction";
(3) The organization of this paper is fine.

Experimental design

(1) The LASSO-LSTM algorithm is proposed in this paper to predict the stock price movement. Please explain the complexity and characteristic of the proposed algorithm. Why this algorithm is most suitable for the research questions?
(2) In figure 3, we know that in the financial statement, the unstructured information is difficult to process, how do you deal with these unstructutured data, in what methods or function? Please detailed explain.
(3) In table 3, do the selected indicators have any relationships? Are they independent indicators? In financial statements, there are complex relationships among the financial elements, if the correlations of the selected elements are very high, the elements/indicators can be merged.

Validity of the findings

(1) What is the advantage of your proposed algorithm? I can hardly see the difference in Figure 7. Please enlarge the results pictures.
(2) What is the theoretical contribution to the previous related algorithms and models? Please describe in details.

Additional comments

(1) the topic is interesting since the authors using intelligent methods to solve the questions in financial management and stock price prediction.
(2) the proposed algorithms which are the core content of this paper should be completed in theoretical contribution, model formulation, data processing in unstructured data, etc.
(3) Results should be compared in different dimentions.

---

## Round 0.2 · accepted · Accept

This is interesting work. Looking forward to your future work!

Reviewer 1 ·

Basic reporting

Clear and unambiguous, professional English used throughout.

Experimental design

Original primary research within Aims and Scope of the journal.

Validity of the findings

All underlying data have been provided; they are robust, statistically sound, & controlled.

Additional comments

The quality of this revised version is obviously greatly improved, several key questions have been answered in detail, the format is more standard, and the text is more accurate. It is recommended to accept this paper

Reviewer 2 ·

Basic reporting

1. In the revised version of manuscript, the presentation is fine and clear.
2. The authors revised the literature review in Introduction, but the related work seems have little update in new researches.
3. The organization of this manuscript is fine, the figures, tables are presented well.
4. The authors have re-described the algorithm in well structures.

Experimental design

1. The manuscript can meet the aims and scope of the submitted journal.
2. Research questions are well discussed in the revised manuscript.
3. Methods are described with sufficient details and information to replicate.

Validity of the findings

The findings are now clearly illustrated.
I suggest that the author should highlight your contributions in both theoretical and practical in Conclusions, and Implications should be presented.